# OCCAM: Online Continuous Controller Adaptation with Meta-Learned Models

**Hersh Sanghvi, Spencer Folk, Camillo Jose Taylor**
School of Engineering and Applied Science
University of Pennsylvania
{hsanghvi,sfolk,cjtaylor}@seas.upenn.edu

**Abstract:** Control tuning and adaptation present a significant challenge to the usage of robots in diverse environments. It is often nontrivial to find a single set of control parameters by hand that work well across the broad array of environments and conditions that a robot might encounter. Automated adaptation approaches must utilize prior knowledge about the system while adapting to significant domain shifts to find new control parameters quickly. In this work, we present a general framework for online controller adaptation that deals with these challenges. We combine meta-learning with Bayesian recursive estimation to learn prior predictive models of system performance that quickly adapt to online data, even when there is significant domain shift. These predictive models can be used as cost functions within efficient sampling-based optimization routines to find new control parameters online that maximize system performance. Our framework is powerful and flexible enough to adapt controllers for four diverse systems: a simulated race car, a simulated quadrupedal robot, and a simulated and physical quadrotor. The video and code can be found at https://hersh500.github.io/occam.

**Keywords:** Controller Adaptation, Robot Model Learning, Meta-Learning

## 1 Introduction

Robust and highly-performant control policies are critical for the successful application of robots in diverse environments. All control algorithms for robots have parameters that must be carefully tuned for good performance, but whose optimal values are generally not obvious *a priori* and are not easily computed. In practice, these parameters are often tuned based on the designer's intuition and using heuristics that do not align with the actual downstream task of the robot–for instance, gain tuning a quadrotor using the step response. This problem is often exacerbated by the *sim-to-real* gap [1], requiring designers to fine-tune control parameters directly on the real system. Furthermore, the optimal control parameters can vary over time if the robot's physical attributes change or the robot enters new environments. Modern data-driven approaches that adapt to these changes usually perform online system identification (potentially in a latent space) for optimization-based control or policy learning [2, 3, 4, 5]. These adaptive methods have shown strong results, though for many of these approaches, it is difficult to tune their behaviors after training, and their generalizability to novel domains is unclear.

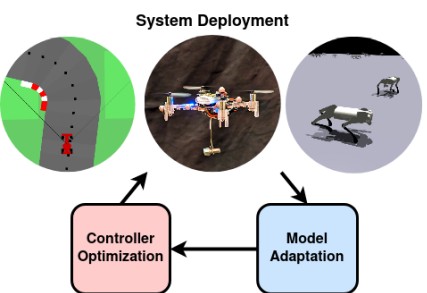

Figure 1: We demonstrate a flexible method for controller optimization based on online adaptation of meta-learned models on diverse robots.

8th Conference on Robot Learning (CoRL 2024), Munich, Germany.

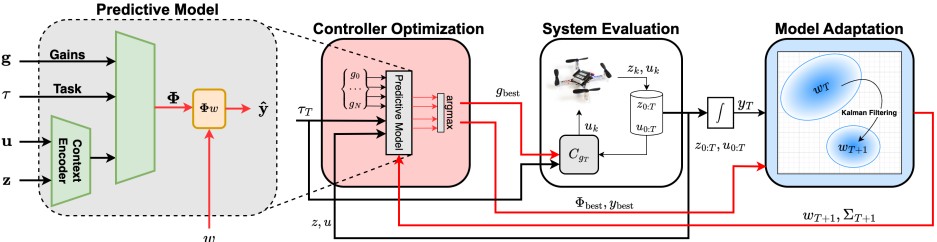

Figure 2: An overview of our method and predictive model $\hat{f}$. Given previous sensor measurements and inputs from the system, the Optimization phase uses the prediction model to search for the best gain to try next. This gain is sent to the real system for the Evaluation phase, during which sensor measurements and performance measures are collected. The Adaptation phase computes new weights to update the prediction model with the information gathered during the evaluation. Arrows in red indicate predicted and estimated quantities, while arrows in black indicate signals from the actual system.

In this paper we present OCCAM, a framework for online controller adaptation to new environments using adaptable learned models of performance metrics. Our main contribution is applying meta-learning and online Bayesian regression to train predictive models on simulated data that provide a strong prior on controller performance and are readily adapted to data gathered during online operation. We show that we can use these models as cost functions to search for optimal control parameters in new environments unseen during training. One of the major advantages is the flexibility of our framework: it can easily be applied to any system with a parameterized controller. We demonstrate that our method performs online controller adaptation on a simulated race car equipped with a nonlinear control law, a simulated and physical quadrotor micro-aerial-vehicle (MAV) with a model-based geometric controller, and a simulated quadruped with a neural-network locomotion policy.

## 2   Related Work

**Automated Controller Tuning:** Approaches for controller tuning often use Bayesian Optimization (BO) with Gaussian Process (GP) surrogate models [6, 7, 8, 9], probabilistic sampling-based approaches [10, 11, 12], or gradient descent through the controller and dynamics [13]. Prior data can be incorporated into BO by simply adding datapoints to the GP or specifying a prior mean function. [14, 15] add prior datapoints to a GP to find control policies for damaged robots online; [16, 17] instead use offline data to learn fixed kernel functions that make BO more efficient. These methods can degrade if the kernel or prior becomes inaccurate as the true underlying function changes. Our method is instead trained to accommodate these domain shifts.

**Adaptation in Data-Driven Control:** A closely-related line of works combines Reinforcement Learning (RL) for learning low-level control policies with system identification. Short-term histories, or *contexts*, of states and actions are encoded as latent variables to condition either model-free policies [5, 18, 19, 20, 21] or dynamics models [22], allowing the model to modulate its behavior in different environments. [23, 24, 25, 26] instead learn a repertoire of diverse policies for varying environments. The deep context encoders common in these works might not be accurate when the test system lies outside training distribution. A key idea in our method is to explicitly adapt the model to the observed performance online.

**Meta-Learning for Robotics:** Meta-learning aims to deal with the lack of adaptability of learned models by incorporating the adaptation procedure into the training loss to train models that are well-suited to efficiently fit new data. [27, 28, 29] use meta-learning to train adaptable mean, kernel, and acquisition functions for BO with applications to controller tuning. [30] use Bayesian regression to meta-learn dynamics models that can be adapted to new dynamics at test-time. [21, 31, 32, 33] employ meta-learning to train low-level control policies with RL that can be adapted quickly to new

contexts. In this work, we alleviate overfitting issues common to these meta-learning methods [34] by adapting our model in a low-dimensional space and training it to characterize the uncertainty in its predictions.

**Adaptive Control:** Adaptive control generally seeks to augment the control inputs to a system to cancel out disturbances and align the system's closed-loop dynamics to those of a desired reference model [35]. Both *indirect* and *direct* adaptive control approaches such as $\mathcal{L}$1-Adaptive control, online model learning, and Model Reference Adaptive Control have successfully been applied to robotic aerial vehicles [36, 37, 38, 39, 40], manipulators [41], and bipedal robots [3, 42]. Poor values of the tunable adaptive gains introduced by adaptive control can render the controller unstable [43]. It also may not be possible for the actual system to track the chosen reference model, and adaptive control generally requires strong assumptions about model structure and differentiability [38, 39], which our method does not require.

## 3  Problem Statement

In this work we address the problem of continuously adapting a controller using a combination of offline simulation data and data gathered from the robot online. The goal of the OCCAM procedure is then to find the control gains at iteration $k$, $g_k$, that maximize the reward over some upcoming time horizon $\Delta T$ that is chosen by the user:

$$g^* = \arg\max_g J(y_{k+\Delta T}) = \arg\max_g J(f_{\theta_k}(g_k, z_{k:k+\Delta T}, u_{k:k+\Delta T}, \tau_k)) \tag{1}$$

Where we let $z_k \in \mathbb{R}^{N_z}$ denote a vector containing the sensor observations from the system at the $k$'th discrete timestep and $u_k \in \mathbb{R}^{N_u}$ denote the control inputs applied to the system by the controller. $y_{k+\Delta T} \in \mathbb{R}^{N_y}$ denotes a vector of performance measures, such as tracking error, control effort, or speed, computed as a fixed function of the control gains, measurements and control history over the horizon $\Delta T$: $y_{k+\Delta T} = f(g_k, z_{k:k+\Delta T}, u_{k:k+\Delta T}, \tau_k)$. In addition, we specify a reward function, $J$ that maps the vector of performance measures onto a single scalar value: $J(y) \in \mathbb{R}$.

The dynamics of our system are governed by a set of unobserved intrinsic and extrinsic system parameters such as inertial parameters or friction coefficients. We refer to these system parameters collectively as $\theta_k$. When generating data in simulation the designer would typically perform domain randomization by sampling possible values of $\theta$ to explore the behavior of the system in different regimes.

We are also given a control policy, $C$ (e.g. PID, MPC, etc.) that computes control inputs from previous measurements: $u_{k+1} = C_{g_k}(\tau_k, z_{0:k}, u_{0:k})$. $C$ is parameterized by tunable gains $g_k \in \mathbb{R}^{N_g}$, and $\tau_k$, which is the control objective of the controller derived from the robot's task. For example, $\tau_k$ could denote an encoding of the upcoming trajectory that we want a mobile robot to follow.

## 4  Method

### 4.1  Modeling Controller Performance

The foundation of OCCAM is a predictive model $\hat{f}$ which is designed to predict future performance measures as a function of the chosen control parameters. The structure of this prediction model is shown in Figure 2 (left). The recent measurements and control inputs (from a sliding window of size $H$), $z_{k-H:k}, u_{k-H:k}$, are fed into a shallow fully connected network that produces a context encoding [5] that captures relevant, observable aspects of the system. This encoded result is combined with an encoding of the current task, $\tau_k$ and the proposed control parameters, $g_k$ and fed into another fully connected network.

To quickly adapt to new data received from the robot during online deployment, the second fully connected network employs a last linear layer structure [30, 44] wherein the network outputs a

matrix $\Phi \in \mathbb{R}^{N_y \times N_b}$ that is combined with a weight vector $w_t \in \mathbb{R}^{N_b}$ to produce the final prediction: $\hat{y}_t = \Phi_t w_t$. We can think of the columns of the matrix $\Phi = [\phi_1, ...\phi_{N_b}]$ as a set of basis functions weighted by $w_t$. Here we use $t$ as an index instead of $k$ to denote that this adaptation occurs at a different rate than the low-level control and sensing, with $\Delta t > \Delta k$. Due to uncertainties in the model and measurements, we model $w_t$ as a normally distributed random variable: $w_t \sim \mathcal{N}(\mu_t, \Sigma_t)$. Because $\hat{y}$ is a linear transformation of $w$, we obtain the following distribution for it: $\hat{y}_t \sim \mathcal{N}(\Phi_t \mu_t, \ \Phi_t \Sigma_t \Phi_t^T)$. We use a Kalman filter with identity dynamics to recursively estimate $w$ as new data, $(\Phi_t, y_t)$, is obtained online from the system at test time:

$$
\begin{aligned}
\bar{\Sigma}_{t+1} &= \Sigma_t + Q \\
K &= \bar{\Sigma}_{t+1} \Phi_{t+1} \left( \Phi_{t+1} \bar{\Sigma}_{t+1} \Phi_{t+1}^T + R \right)^{-1} \\
w_{t+1} &= w_t + K(y_{t+1} - \Phi_{t+1} w_t) \\
\Sigma_{t+1} &= (I - K\Phi_{t+1}) \bar{\Sigma}_{t+1}
\end{aligned}
\tag{2}
$$

Informally adopting standard Kalman Filter terminology, $Q$ is a positive definite matrix that scales with our confidence in keeping $w$ constant, while $R$ captures the error in our model, $\Phi$, and the noise in the measurements of $y$. More information on this formulation can be found in the appendix.

## 4.2 Meta-Training

While training our model on offline data, our goal is to produce learned basis functions that capture the variation in performance we observe across different system parameters, $\theta$. Instead of estimating $\theta$ directly, our approach adapts in the latent space of $w$. To achieve this, after pretraining our network for a few epochs using standard gradient descent, we switch to gradient-based meta-learning [31], which consists of a *base learner* which adapts the base model's initial parameters to data from different "scenarios", and an *outer loop* which adjusts the initial parameters of the base model using the loss achieved by the base learner after adaptation. Our base learner is the recursive Kalman update (2) applied sequentially to $n$ random datapoints gathered from a system with

---

**Algorithm 1:** Meta-Learning with Kalman Filter Base Learner

**Input:** Training dataset $\mathcal{D}$, network parameters $\psi$, $w_{\text{pre}}$
**Output:** Trained parameters $\psi, w_0, \Sigma_0, Q, R$
Initialize $w_0 = w_{\text{pre}}$ ;
Initialize $\Sigma_0, Q, R = I$ ;
**foreach** *training epoch* **do**
    **foreach** *parameter dataset* $D_\theta = (G, Z, U, \tau, Y) \in \mathcal{D}$ **do**
        Sample subset $\mathcal{D}_{\text{train}}$ of random size $n$ from $\mathcal{D}_\theta$;
        Initialize $w_0' = w_0, \Sigma_0' = \Sigma_0$ ;
        **foreach** *datapoint* $(g_i, z_i, u_i, \tau_i, y_i)$ *in* $\mathcal{D}_{\text{train}}$ **do**
            | Compute $w_i', \Sigma_i'$ using (2)
        **end**
        Compute $\mathcal{L} = \|\Phi_\psi(G, Z, U, \tau)w_n' - Y\|^2$ ;
        **foreach** $p \in \{\psi, w_0, \Sigma_0, Q, R\}$ **do**
            | $p \leftarrow p - \alpha \nabla_p \mathcal{L}$ ;
        **end**
    **end**
**end**

---

new $\theta$. Because every computation in the Kalman Filter is fully differentiable, once the final weights $w_n$ are computed, the gradient of the predictive loss can be backpropagated through the full computation graph to update the neural network parameters and the Kalman filter parameters $w_0, \Sigma_0, Q, R$ using standard automatic differentiation tools. Algorithm 1 shows how we use the standard meta-learning loops with our Kalman Filter base learner as the inner loop. Meta-learning $w_0$ and $\Sigma_0$ is similar in philosophy to Harrison et al. [30] and Bertinetto et al. [45], which also learn the prior distribution for the last-layer weights. However, they do not consider a Kalman Filter formulation for regression, and thus do not include process and measurement covariance matrices $Q$ and $R$.

## 4.3 Controller Optimization With Learned Model

When running OCCAM online, we want to use the learned prediction model to solve (1) while also using our knowledge of the uncertainty on the model's predictions. More specifically, at regular

intervals OCCAM seeks to find a choice of control parameters that maximizes the uncertainty-aware reward:

$$g^* = \arg\max_g \mu(J(\hat{y}_{k+\Delta T})) + \beta\sigma(J(\hat{y}_{k+\Delta T}))$$

where $\hat{y}_{k+\Delta T} = \Phi(g_k, \tau, z, u)w_k$ is the output of the prediction model, and $\beta$ is a scalar penalty that controls the confidence bound. While many optimizers could be used, we employ a particle filter-based random search in our evaluation settings. In each search iteration, we generate a fixed number of random controller gains from a uniform distribution $G_r = \mathcal{U}(g_{\min}, g_{\max})$. We also propagate previous samples by applying $N_r$ random perturbations to the optimal gains found in the $T$ previous trials, generating $T \times N_r$ search samples $G_e$. The full set of samples is thus $G = G_r \cup G_e$. Each sample gain is concatenated with the upcoming command, $\tau_k$ and system history, $z_{k-H:k}, u_{k-H:k}$, and passed through the network to obtain the expected reward. After evaluating all samples, we pick the sample with the highest expected reward. This optimization approach introduces minimal hyperparameters and is computationally efficient; all search samples can be collated into a single batch and evaluated with a single forward pass through the network.

## 5 Experimental Procedures

### 5.1 Simulated Evaluation Platforms

To show the flexibility of our framework, we evaluate it on three distinct simulated and one real robotic platform, described below. More details on each system and its accompanying controllers are provided in the supplementary material.

**2D Race Car with Nonlinear Controller:** Our first simulated robotic system is a 2-dimensional car racing around procedurally generated race tracks, modified from OpenAI's gym environment [46]. The car has unknown mass, engine power, and tire friction as $\theta$, with an encoding of the race track shape as task $\tau$. OCCAM's goal is to continually optimize the six-dimensional control gains $g$ of a nonlinear PD steering controller that keeps the car on the centerline of the track, and a proportional acceleration controller that accelerates and brakes the car based on tunable speed thresholds. The speed thresholds improve performance but make the controller nonlinear and nondifferentiable. The reward function is a linear combination of the performance measures: inverse average lateral tracking error, inverse average wheelslip, and average speed over the upcoming track segment. OCCAM adapts and selects new gains every time the car traverses $\frac{1}{3}$ of a lap.

**Quadrotor with Geometric Controller:** Our second simulated platform is a quadrotor MAV equipped with a geometric trajectory tracking controller defined on SE(3) [47], implemented in RotorPy [48]. The quadrotor has five unknown system parameters $\theta$ which are the quadrotor's true mass, principal moments of inertia, and thrust coefficient. The controller computes a feedforward motor speed command based on reference trajectory $\tau$ using the quadrotor's nominal parameters (centered around those of the Crazyflie platform [49]) and a feedback command to correct tracking errors. The controller is parameterized by eight PD gains on the position and attitude. The reward function is a linear combination of the inverse average positional tracking error, inverse average yaw tracking error, inverse average pitch and roll, and inverse average commanded thrust over the episode. The adaptation frequency $\Delta T$ is 4 seconds. We also evaluate our framework on a physical Crazyflie with the same controller and performance measures. Details of the physical experiments are described in Section 6.2.

**Quadrupedal Robot with Learned Locomotion Policy:** Our third simulated robotic platform is a quadrupedal robot equipped with a static pretrained locomotion policy $\pi$ trained using model-free RL [26]. $\pi$ outputs joint angles at 50Hz such that the torso of the robot follows a velocity twist command $\tau_k = (\dot{x}_{\text{des}}, \dot{y}_{\text{des}}, \dot{\omega}_{\text{des}})$. We treat $\pi$ as our controller $C$ for this system. Although $\pi$ is parameterized by a deep neural network, it is also conditioned on an additional, eight-dimensional command $g_k$ that allows the user to specify high-level desired behaviors that the policy should follow: stepping frequency, body height, footswing height, stance width, and three discrete variables

which jointly specify the quadrupedal gait. We treat $g_k$ as the controller parameters to be tuned automatically based on the quadruped's randomized parameters and the twist command. The unknown system parameters $\theta_k$ for the quadruped are added mass payloads to the robot base, motor strengths, and the friction and restitution coefficients of the terrain. The adaptation frequency $\Delta T$ is 3 seconds.

## 5.2 Model Training and Testing

To generate training data from each simulated platform, we randomly sample $N$ parameter vectors $\theta_{0:N}$. For each $\theta_i$, we sample $N_B$ random samples of $g$ and $\tau$, drive the system to a random initial dynamical state to collect history $z, u$, and finally roll-out the system with controller $g$ to gather performance metrics $y$. We train a separate basis-function network for each system.

We test each method on robots with system parameters sampled from outside of the range of the training dataset, testing the ability of each method to extrapolate outside of the training data. We evaluate methods on 15 out-of-distribution system parameters on multiple different tasks (racetracks for the racecar, ellipsoidal trajectories for the quadrotor, and twist commands for the quadruped) and 8 random seeds. For each platform, we fix the reward functions $J$ and uncertainty penalty $\beta$. Each test consists of multiple iterations of the "Optimize, Evaluate, Adapt" procedure shown in Figure 2. Because we run each test in an online setting with no resets between timesteps, if a method selects dangerous gains that cause a crash, the reward for every subsequent timestep of that evaluation is set to zero. More details on dataset sizes, model architecture, and training and testing parameter ranges are provided in the supplementary material.

## 5.3 Baselines

We compare our method on each test system against three baselines. The first baseline is Reptile [50], a first-order version of the common gradient-based meta-learning method MAML [31]. Reptile uses minibatched gradient descent as the base learner and it only outputs a single point prediction of $y$. For our second baseline, we compare against a GP model whose kernel function is the composition of our network trained without meta-learning with an RBF kernel, mirroring [17, 29]. We also compare against control parameters that were hand-tuned by experts for robust and conservative performance across a variety of commands and system parameters. In our quadrotor example, we additionally consider an $\mathcal{L}1$-Adaptive control [36] baseline adapted from [37], which augments the nominal control with an additional term based on an estimated disturbance vector.

To evaluate the usefulness of adapting the prediction model, we perform an ablation where we do not adapt $w$ from trial to trial and only use $w_0$ and $\Sigma_0$ (`context-only`). This ablation mimics methods that directly estimate latent system parameters from state and action histories, such as Kumar et al. [5], since the prediction network only uses the history context encoder and does not utilize our last-layer weight adaptation. To evaluate the usefulness of meta-training, we also run an ablation where last-layer weight adaptation occurs, but the network is trained without our meta-learning procedure (`no-meta`).

## 6 Results

In Table 1 we report the crash rate and final average reward achieved by each method. To show the advantage of each method over simply using the expert-tuned nominal gains, in Figure 3 we plot the difference between each method's reward and the nominal gain's reward over time in each test. We report the raw performance metrics achieved by each method in the supplementary material.

OCCAM outperforms both ablations and all baselines on all benchmarks, demonstrating the utility of both meta-learning basis functions and weight adaptation. Adaptation with our method also occurs within a few timesteps, which corresponds to 10-20 seconds of data on each system. The quadruped environment is challenging in particular, as OCCAM must predict the performance of an expressive RL policy; nonetheless, it finds behavior parameters that improve the policy's performance. The quadrotor also presents unique challenges, as the parameters of the quadrotors in the test

Table 1: Average Final Reward and Crash Rate on Robotic Systems

| Method | Race Car | | Quadrotor | | Quadruped | |
|---|---|---|---|---|---|---|
| | Avg Final Rwd (↑) | Crash % (↓) | Avg Final Rwd (↑) | Crash % (↓) | Avg Final Rwd (↑) | Crash % (↓) |
| Nominal | $0.25 \pm 0$ | **8** | $0.72 \pm 0.24$ | 33.8 | $0.69 \pm 0.08$ | 8.8 |
| LK-GP | $0.29 \pm 0.14$ | 24.2 | $1.11 \pm 0.34$ | 29.9 | $0.74 \pm 0.07$ | 6.6 |
| Reptile [50] | $0.29 \pm 0.25$ | 14.2 | $1.19 \pm 0.37$ | 33.8 | $0.70 \pm 0.09$ | 8.1 |
| $\mathcal{L}$1-Adaptive [36] | - | - | $0.61 \pm 0.25$ | 33.6 | - | - |
| OCCAM (no-meta) | $0.1 \pm 0.16$ | 45.9 | $1.05 \pm 0.42$ | 32.8 | $0.73 \pm 0.09$ | 8.1 |
| OCCAM (context-only) | $0.22 \pm 0.16$ | 37.3 | $1.08 \pm 0.38$ | 27.3 | $0.73 \pm 0.07$ | 6.7 |
| **OCCAM (Ours)** | $\mathbf{0.40 \pm 0.17}$ | 13.4 | $\mathbf{1.40 \pm 0.30}$ | **26.3** | $\mathbf{0.75 \pm 0.07}$ | **6.3** |

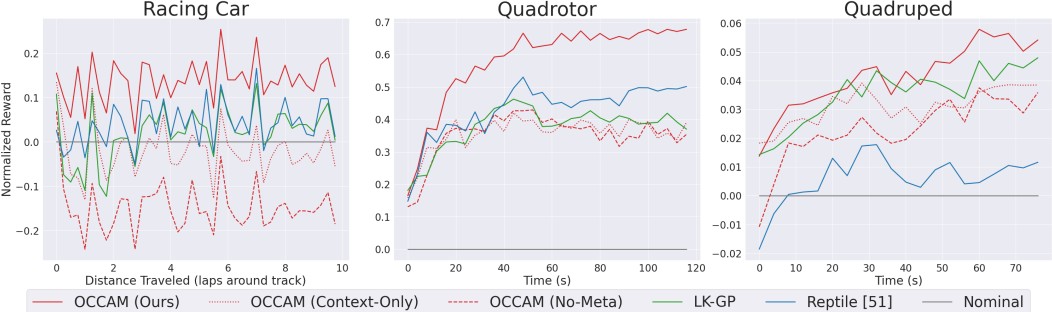

(a) Plots of performance reward value vs time on robotic systems (Higher is Better)

Figure 3: Time vs. Reward curves on all test systems. Our method shows robust prior performance and adaptation across all tests. For the robotic systems, normalized reward is computed by standardizing rewards according to the training dataset statistics for that system and subsequently subtracting the nominal controller's reward.

set can vary from $\frac{1}{3}\times$ to $3\times$ the nominal parameters. In our problem setup, this must be compensated only by tuning PD feedback gains (the controller does not have an integral term). These extreme parametric errors also cause instability and poor tracking in the $\mathcal{L}$1-adaptive controller. Meanwhile, our method finds gains that result in the lowest crash rate and nearly a 50% reduction in tracking error compared to the nominal gains and $\mathcal{L}$1-Adaptive controller. Both Reptile and LK-GP saturate early in the evaluations. Reptile tends to overfit to the training set and online training data [34] and gets stuck in a loop of selecting the same gains and repeatedly fitting to them.

In the supplementary material, we additionally show that OCCAM also performs well on common global function optimization benchmarks, makes interpretable optimizations to the gains as system parameters change, and learns a structured latent weight space.

## 6.1 Weight Adaptation Enables Out-of-Distribution Generalization

To further demonstrate the necessity of model adaptation for controller tuning in *out-of-distribution* scenarios, we also evaluate each baseline on test scenarios that lie *within* the training distribution, and present a table of these results in the supplement. In these test cases, the `context-only` baseline performs comparably to our full method. Therefore, its performance drop in the out-of-distribution scenarios suggests that the learned context encoder alone is not sufficient to achieve generalization. Similarly, our method outperforms LK-GP because the usefulness of the latent space of the kernel network degrades as we move out of its training distribution.

To emphasize OCCAM's generalizability, we also use OCCAM to tune the controller gains of a simulated Crazyflie to minimize tracking error in three different wind conditions up to 0.5 m/s, when wind was **not encountered at all during training**. OCCAM is able to achieve an average tracking error of under 7cm, outperforming all baselines including $\mathcal{L}$1-Adaptive control (10cm), shown in the supplementary material. Meanwhile, `context-only` achieves the worst tracking performance (30cm).

## 6.2 OCCAM crosses the Sim-to-Real Gap

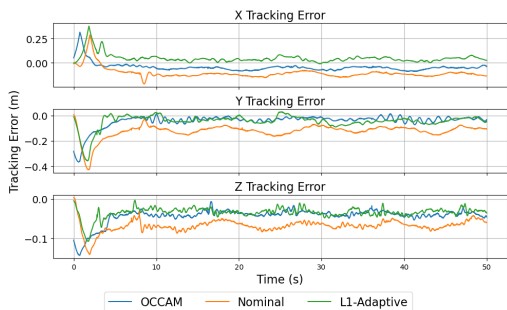

Figure 4: Positional tracking error results on physical Crazyflie quadrotor following a 3-dimensional ellipsoidal reference trajectory.

Finally, we apply OCCAM to tune the gains of a physical Crazyflie quadrotor. The controller for the physical Crazyflie runs on a Robot Operating System (ROS) node running on a laptop with no GPU and a 1.6Ghz Intel i5 processor. The ROS node receives the quadrotor's position through a Vicon motion-capture system[1] running at 100Hz and computes a desired collective thrust and quaternion attitude command for the quadrotor, which the onboard PID controller then converts into motor speeds. We train our predictive model on simulation data sampled from both the train and test parameter ranges, and add OCCAM as a lightweight layer that runs every 3 seconds on top of the ROS control node.

Position tracking error results for a three-dimensional ellipsoidal reference trajectory are shown in Figure 4. Control parameters tuned by OCCAM result in lower tracking error than the expert-tuned nominal controller and achieve comparable tracking error to the $\mathcal{L}1$-Adaptive controller. We also run additional experiments where a 5 gram mass is added to the Crazyflie (see Figure 1), which represents a 20% increase from the its nominal mass. Our method is able to find gains that decrease the quadrotor's tracking error in the z-axis by 54% compared to the nominal controller and by 17% compared to the $\mathcal{L}1$-Adaptive controller. Figures of tracking errors from these physical Crazyflie experiments are shown in the supplementary material.

## 7 Discussion and Limitations

Our results demonstrate a single algorithm that optimizes controllers on a diverse array of robotic platforms using simple search techniques. Importantly OCCAM obviates the need for manual gain tuning that typically requires domain expertise, and instead provides automatic controller adaptation based on task-relevant metrics. OCCAM is also compatible with any controller with tunable parameters, allowing straightforward application to controllers that are nonlinear, discontinuous, and not analytic. As shown in our Crazyflie experiment, OCCAM's models and optimization routine are lightweight enough to run in realtime without a GPU. We expect OCCAM to perform best on systems with a well-defined and relatively smooth mapping between control gains and performance; this and further assumptions are discussed in the appendix.

In this work, we tested OCCAM on commonly used controllers with relatively low-dimensional gain spaces, though one might also wish to tune a higher-dimensional policy, such as a neural network. Although the policy could be conditioned on a low-dimensional behavior vector, one could also treat the weights of either the entire neural network or its last few layers [51, 52] as the controller gains and use a more powerful optimizer.

Beyond this, there are a few areas in which our method could be improved. If the stochastic optimizer makes a poor initial choice and the uncertainty penalty is poorly tuned, OCCAM sometimes settles early in a suboptimal local maximum. For some systems, this incorrect choice of gains can cause crashes. Incorporating a more sophisticated optimizer and exploration strategy would be interesting future work. Lastly, in this work we do not rigorously evaluate *how far* the generalizability of our method applies, or changing the reward function $J$ online. Future work includes addressing these limitations, along with demonstrating our method on broader classes of systems.

---

[1]https://www.vicon.com/

**Acknowledgments**

The authors would like to thank the reviewers, who gave helpful feedback during the discussion phase. This work was supported by NSF grant CCF-2112665 (TILOS).

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

# A    Table of Notation

Table 1: Table of Model Elements

| | |
|---|---|
| $z_k$ | Vector of sensed values |
| $z_{0:k}$ | History of sensed values to date |
| $u_k$ | Vector of control inputs |
| $u_{0:k}$ | History of control inputs to date |
| $y_{k+\Delta T}$ | Performance metrics over time $k$ to $k + \Delta T$ |
| $J$ | Reward function maps performance metrics to a scalar |
| $\theta_k$ | System parameters, intrinsic and extrinsic (unknown) |
| $\tau_k$ | Control objective derived from current task |
| $g_k$ | Vector of control parameters |
| $C$ | Control law $u_k = C(g_k, \tau_k, z_{0:k}, u_{0:k})$ (grey-box) |
| $\Delta T$ | Frequency at which OCCAM adapts and computes new gains |

# B    Discussion on Kalman Filter Formulation

In this section, we will discuss the aspects of our formulation that lead to the choice of the Kalman Filter as our online estimator. First, we choose a linear output layer of our neural network: $y = \Phi(g, z, u, \tau)w$, which has been shown to improve model generalization in meta-learning settings [1, 2]. Then, given data pairs $(\Phi_t, y_t)$ received in a stream from the robot, solving for the optimal mapping s.t. $\Phi w = y$ is an online linear regression problem. However, there is significant uncertainty in the regression problem arising from our lack of knowledge of $\theta$, combined with epistemic uncertainty in the model and environmental noise. Therefore, we would like to estimate the posterior distribution of $w$ to characterize the uncertainty in the model, especially for the purposes of downstream decision making using the network's predictions. This estimate also needs to be able to vary with time, since the environment parameters $\theta$ could slowly change as the robot moves into a new environment. Given the choice of a linear measurement model and the additional factors above, this strongly resembles the time-varying parameter estimation problem, to which a linear Kalman Filter with identity dynamics is commonly applied. Representing $w$ as a Gaussian random variable enables analytic computation of the posterior and keeps the filtering procedure differentiable, which enables our meta-learning pipeline.

# C    Applicability of OCCAM

Although our experiments show that OCCAM can be applied to a wide variety of systems, we do not yet have a precise theoretical characterization of OCCAM's performance. However, we do make a few implicit assumptions about the kinds of systems that OCCAM would work well on.

1. For OCCAM to work well, the closed loop dynamics of the system should allow a well-defined mapping between control gains and performance. This mapping does not have to be known, can vary as the system parameters vary, and can be stochastic. However, if it is too noisy or nonsmooth, then the performance of the predictive model will degrade. For example, if the closed loop dynamics are chaotic, then the effect of control gains on system performance might be unpredictable. Although we make no assumptions about the shape of the mapping, we expect OCCAM to work best if the mapping is relatively smooth.

2. We also assume that the offline dataset (generated through simulation or from previous experience) captures the important characteristics of how gains affect performance and how that mapping changes as the environment changes. Although our method can adapt to a significant domain gap (as demonstrated by our experiments), if the domain gap is too large then the basis functions and prior weights learned from offline data will not be useful and our method will take many iterations to adapt.

3. The closed-loop system should be stable enough that a suboptimal set of control gains (such as those deployed in the early iterations of adaptation) will not cause an immediate crash.

4. Our use of a sampling-based optimization method limits the dimensionality of gain spaces that our method can handle. However, we could use a more powerful optimizer to overcome this, and many model-based control methods of interest (PID, MPC, LQR, etc.) do have gain spaces that are low-dimensional.

## D    Details on Simulated Platforms

In this section we provide additional details about each of our simulated evaluation platforms, including two benchmark functions which are commonly used to test global functional optimization algorithms.

### D.1    Benchmark Functions

We first validate our method on randomized variations of two common global optimization benchmark functions [3]. The first is the Branin function, which has a 2D input space and 1D output space:

$$f(x) = a(x_2 - bx_1^2 + cx_1 - r)^2 + s(1 - t)\cos(x_1) + s$$

We treat as system parameters the six constants that parameterize the shape of the Branin function: $\theta = [a,\, b,\, c,\, s,\, t,\, r]$.

The second is the Hartmann function, which has a 6D input space and 1D output space:

$$f(x) = -\sum_{i}^{4} \theta_i \exp\left(-\sum_{j=1}^{6} A_{ij}(x_j - P_{ij})^2\right)$$

Where $A$ and $P$ are constant matrices, and we randomize over the 4-dimensional vector $\theta$ as system parameters.

For these benchmark functions, there are no measured quantities $z_{0:k}$ or control actions $u_{0:k}$. We consider the inputs to the benchmark functions to be the "gains" $g_k$, and the outputs of the functions to be the performance measures $y_k$. Therefore the data tuples for these functions consist of only the inputs $x$ and scalar "metrics" $y = f(x)$. For these functions, the reward function is simply set to the negative of the scalar function values: $J(y) = -y$. Because there is no history context, the `context-only` baseline in these two examples is simply our method without weight adaptation.

For the benchmark functions, we use F-PACOH [4], which is based on training neural networks with regularization to serve as mean and kernel functions in a GP. F-PACOH is ill-suited to our robotic tests due to the high dimensionality of the full input space to the networks, so we use the LK-GP baseline in our robotic experiments instead.

### D.2    2D Race Car

Our first simulated robotic system is a 2-dimensional car racing around a track, modified from the OpenAI Gym "Car Racing" environment [5]. The environment models a powerful rear-wheel-drive car with sliding friction, making control nontrivial while trying to maximize speed on track. The system has three control inputs: $u_k = [u_s, u_g, u_b]$. The controller $C$ of the car consists of a nonlinear proportional-plus-derivative (PD) controller that computes steering input $u_s$ to steer the car towards the centerline of the track and a simple control law that accelerates the car by force $u_g$ on straightaways up to a maximum speed, or brakes the car by force $u_b$ for corners above a certain curvature threshold. The tunable maximum speed and curvature braking thresholds induce saturation and branching behavior in the controller. The sensor measurements of this system are $z_k = [v, \omega_k, e_{\text{lat}}]$, where $v$ is the forward velocity of the car, $\omega_k$ is the angular velocity, and $e_{\text{lat}}$ is the lateral distance between the car and the track centerline. The controller $C$ uses $z_k$ and an

estimate for track curvature, $c$, derived from a vector of upcoming track waypoints $\tau_k$ to compute $u_k$ as follows:

$$u_s = k_{ps} e_{\text{lat}} + k_{ds} \dot{e}_{\text{lat}}$$

$$u_g = \begin{cases} k_{pg}, & \text{if } v \leq v_{\text{max}} \\ 0, & \text{otherwise} \end{cases}$$

$$u_b = \begin{cases} k_{pb}, & \text{if } c \geq c_{\text{thresh}} \\ 0, & \text{otherwise} \end{cases}$$

The tunable parameters of this controller are

$$g = [k_{ps} \ k_{ds} \ k_{pg} \ k_{pb} \ v_{\text{max}} \ c_{\text{thresh}}]$$

The racing car environment has three unknown system parameters $\theta = [m, \ p, \ \mu]$, which are respectively the mass of the car, the car's engine power, and the friction between the tires and track.

For the racing car, the evaluation function computes the vector of performance metrics

$$y_{k:k+\Delta T} = \frac{1}{\Delta T} \left[ \frac{1}{1 + \sum_i e_{\text{lat}i}}, \frac{1}{1 + \sum w_i}, \sum v_i \right]$$

These are respectively the inverse average lateral tracking error, inverse of total number of timesteps during which a wheel was slipping, and average velocity over a fixed evaluation horizon. We invert tracking error and wheelslip since, in general, they ought to be minimized. In this case, the evaluation horizon is not a fixed $\Delta T$ but instead is however long it takes for the car to traverse a fixed distance on track. For online testing of this system, we set the reward function to be a weighted combination of the reward terms: $J(y) = \sum_{j=0}^{3} r_j y[j]$.

The reasoning behind inverting certain performance measures is to focus the model onto gains which have good performance. In this racing car setting, for example, "good performance" equates to low tracking error, low wheelslip, and high speed. At test time, we will only select and deploy gains for which the network predicts good performance. Therefore, our preference is that the network should be very discriminating between two gains that have good performance, i.e. low tracking error and wheelslip. If two gains result in high tracking errors, we do not care about exactly predicting that difference, because we will not deploy those gains. However, during training time, using the Mean Squared Error Loss function has the opposite effect - the network predicting that a gain will have a tracking error of 1m vs 2m will result in a very large gradient, while the network mispredicting by a small amount will have a small effect on the gradient. Inverting the performance measures for which lower values are better has the desired effect of increasing the distance between high performing gains, while compressing the "bad" gains into a small region of the space. And because the inversion is invertible, we can recover the original, interpretable metrics if necessary.

### D.3 Quadrotor with Model-Based Controller

Our second simulated platform is a quadrotor MAV equipped with a geometric trajectory tracking controller defined on SE(3) [6]. This controller takes in a reference trajectory $\tau_k$ defined in the quadrotor's flat output space: position $(p_x, p_y, p_z)$ and yaw. The controller computes a feedforward motor speed command based on $\tau_k$ using the quadrotor's nominal mass, inertial tensor, thrust and drag torque coefficients. It then uses measurements from the quadrotor $z_k = [p_x, \ p_y, \ p_z, \ v_x, \ v_y, \ v_z, \ R]_k$, where $R$ is the rotation matrix representation of attitude, to compute feedback commands to correct tracking errors. The controller is parameterized by PD gains on the 3D position and PD gains on the attitude: $g_k = [k_x, k_v, k_R, k_\Omega]$ (following the convention given by [6]). The quadrotor has five unknown system parameters which are the quadrotor's mass, principal moments of inertia, and thrust coefficient: $\theta = [m, I_{xx}, I_{yy}, I_{zz}, k_\eta]$. The baseline controller is only aware of the nominal parameters, which are centered around those of the Crazyflie platform [7], and not the actual values. Thus, the feedback gains must be used to compensate for this parametric error. For more detailed information about the quadrotor's dynamics and the controller derivation, see [6].

For this system, the four performance measures $y$ are the inverted average positional tracking error, inverted average yaw tracking error, inverted average pitch and roll, and inverted average commanded thrust over the episode. Following the racing car example, we choose the reward function to be a weighted combination of the terms of $y$: $J(y) = \sum_{j=0}^{4} r_j y[j]$. For this system, we set $\Delta T = 4$ seconds.

For our quadrotor experiments, the commanded trajectories $\tau$ consist of 3-dimensional ellipsoidal trajectories of varying radii and frequencies. Because of the simplicity of these trajectories, we do not have to provide information about $\tau$ as input to the network for this system. We leave the incorporation of more general and complex trajectories to future work. We use RotorPy [8] and its included SE(3) controller for all quadrotor simulations. For this environment, we also evaluate our framework on a physical quadrotor with the same controller and performance measures.

### D.4 Quadrupedal Robot with Learned Locomotion Policy

Our third simulated robotic platform is a quadrupedal robot equipped with a static pretrained locomotion policy $\pi$ trained using model-free RL [9]. $\pi$ outputs joint angles such that the torso of the robot follows a velocity twist command $c_k = (\dot{x}_{\text{des}}, \dot{y}_{\text{des}}, \dot{\omega}_{\text{des}})$. The policy takes as high-dimensional input measurements $z_k$ the joint positions and velocities $q_k, \dot{q}_k$, previous joint angle commands $a_{k-1}$, commands $c_k$, timing reference variables, and estimated base velocity and ground friction. We treat $\pi$ as our controller $C$ for this system.

Although $\pi$ is parameterized by a deep neural network, it is also conditioned on an additional command that allows the user to specify high-level behaviors that the policy should follow:

$$b_k = \left[ \theta_1^{\text{cmd}}, \theta_2^{\text{cmd}}, \theta_3^{\text{cmd}}, f^{\text{cmd}}, h^{\text{cmd}}, h_f^{\text{cmd}}, s^{\text{cmd}} \right]$$

The three terms $\left[ \theta_1^{\text{cmd}}, \theta_2^{\text{cmd}}, \theta_3^{\text{cmd}} \right]$ jointly specify the quadrupedal gait, $f^{\text{cmd}}$ is the commanded stepping frequency, $h^{\text{cmd}}$ is the commanded body height, $h_f^{\text{cmd}}$ is the commanded footswing height, and $s^{\text{cmd}}$ is the commanded stance width. Thus, the policy tries to follow the velocity command $c_k$ while satisfying the behavior constraints. In the original work $b_k$ is a quantity to be selected by a human operator, while in this work we treat $b_k$ as the controller parameters to be tuned automatically based on the quadruped's randomized parameters and the task $c_t$. For details on how the learned policy is trained, see [9].

The randomized system parameters $\theta_k$ for the quadruped are added mass payloads to the robot base, motor strengths, and the friction and restitution coefficients of the terrain. Although the $\pi$ contains an estimator module to regress the ground friction, it does not receive direct observations of any of these parameters.

For use in our method, we input only a reduced-dimension subset of $z_k$ into our prediction model network consisting of the estimated base linear and angular velocities and joint torques applied by the motors.

The four performance measures for the quadruped are the inverted average velocity errors along each axis of the command and inverted total commanded torque over the evaluation horizon. For this system, we set the evaluation horizon $\Delta T = 3$ seconds.

The reward function for the quadruped has the same form as the quadrotor system: $J(y) = \sum_{j=0}^{4} r_j y[j]$. All simulations are done using code and pretrained models from [9] and the Isaac Gym simulator [10].

## E  Model Training and Testing Details

### E.1  Network Pretraining

We observe that if the basis function network $\Phi$ is initialized randomly, the inner-loop adaptation steps used in meta-training can become unstable and cause training to diverge. Other works that

|  | History Size | Encoder Layers | Encoded Dim | Network Layers | Nonlinearity | Basis Size | Phase 1 epochs | Phase 2 epochs |
|---|---|---|---|---|---|---|---|---|
| Branin | - | - | - | [16,16,16] | ReLU | 5 | 50 | 45 |
| Hartmann | - | - | - | [32, 32, 32] | ReLU | 15 | 75 | 45 |
| Racing Car | 25 | [32, 32] | 15 | [32, 32, 32] | ReLU | 5 | 40 | 55 |
| Quadrotor | 25 | [64, 64] | 15 | [64, 64, 64] | ReLU | 15 | 50 | 40 |
| Quadruped | 20 | [64, 64] | 15 | [64, 64, 64] | ReLU | 15 | 50 | 15 |

Table 2: Architecture and Training Hyperparameters for OCCAM Basis Function Network for all tested systems

|  | History Buffer Size | Encoder Layers | Encoded Dim | Network Layers | Nonlinearity | Meta Training Epochs | Inner Loop Steps |
|---|---|---|---|---|---|---|---|
| Branin | - | - | - | [16,16,16] | ReLU | 35 | 10 |
| Hartmann | - | - | - | [32, 32, 32] | ReLU | 70 | 20 |
| Racing Car | 25 | [32, 32] | 15 | [32, 32, 32] | ReLU | 70 | 10 |
| Quadrotor | 25 | [64, 64] | 25 | [64, 64, 32] | ReLU | 25 | 20 |
| Quadruped | 20 | [64, 64] | 15 | [64, 64, 64] | ReLU | 35 | 20 |

Table 3: Architecture and Training Hyperparameters for Reptile baseline for all tested systems

use meta-learning with closed-form inner-loop solvers use pre-trained networks [1, 2] to speed up training and solve this problem. To mimic this approach in our setting, we divide our training process into two phases. In the first phase, the network is trained as an *average* model using standard stochastic gradient descent, with $w_{pre}$ as a *fixed* last layer that does not adapt to tasks. In the second phase, we initialize $w_0 = w_{pre}$ and we switch to meta-training with the Bayesian recursive update as the inner loop optimizer. $\Sigma_0$, $Q_0$, and $R_0$ are initialized before the second training phase as identity matrices. Our `no-meta` ablation uses a network that is trained only with this first phase and not with our meta-learning procedure.

## E.2 Dataset Generation

The datasets for the robotic systems each consist of $N = 1500$ batches of $N_B = 64$ datapoints each. The hyperparameters of each dataset and network are provided in the supplementary material. Note that our method does not require sampling only optimal or high-performing gains to generate data - only random ones. Thus, the dataset for each system consists of $N$ batches of datapoints: $[(g, \tau, z, u, y)_{0:N_B}]_{0:N}$. Each of these batches is used as a "task" for a single inner loop during the meta-training process.

## E.3 Model Architecture

We find that we are able to use small networks to model each system; the networks are all fully-connected networks that consist of 3 hidden layers with fewer than 64 hidden units, outputting between 5-20 bases, indicating that many of the robotic systems that we are interested in controlling can be effectively modeled with a relatively small number of parameters. The exact network layer sizes and training hyperparameters are given in the supplementary material. All models are implemented and trained in PyTorch [11].

Architectural details and training hyperparameters for OCCAM's basis function network, Reptile, and F-PACOH are presented in Tables 2, 3, and 4 respectively. The F-PACOH training hyperparameters were chosen in accordance with experiments conducted in the original paper.

Training and testing parameter ranges for each system evaluated in this work are shown in Tables 5, 6, 7, 8, and 9. For the reward curves and tables shown in the main submission, test system parameters

|  | Network Layers | Num fitting iters | Weight Decay | Prior Factor | Feature Dim |
|---|---|---|---|---|---|
| Branin | [32,32,32] | 2500 | 3e-5 | 0.06 | 5 |
| Hartmann | [32, 32, 32] | 2500 | 0.03 | 0.23 | 6 |

Table 4: Training Details for F-PACOH baseline for all tested systems

| | Training | | Testing | |
|---|---|---|---|---|
| **Parameter** | low | high | low | high |
| a | 0.8 | 1.2 | 0.5 | 1.5 |
| b | 0.11 | 0.13 | 0.1 | 0.15 |
| c | 1.2 | 1.8 | 1 | 2 |
| r | 5.5 | 6.5 | 5 | 7 |
| s | 9 | 11 | 8 | 12 |
| t | 0.035 | 0.045 | 0.03 | 0.05 |

Table 5: Parameter ranges for Branin experiments

| | Training | | Testing | |
|---|---|---|---|---|
| **Parameter** | low | high | low | high |
| $\theta_1$ | 1.0 | 1.5 | 0.5 | 1.5 |
| $\theta_2$ | 1.0 | 1.2 | 0.6 | 1.4 |
| $\theta_3$ | 2.4 | 3.0 | 2.0 | 3.0 |
| $\theta_4$ | 3.0 | 3.4 | 2.8 | 3.6 |

Table 6: Parameter ranges for Hartmann experiments

| | Training | | Testing | |
|---|---|---|---|---|
| **Parameter** | low | high | low | high |
| Size | 0.01 | 0.03 | 0.005 | 0.04 |
| Engine Power | 2.5e4 | 4.5e4 | 2e4 | 5e4 |
| Friction Limit | 250 | 450 | 200 | 500 |

Table 7: Parameter ranges for Racing Car. Note that these quantities are given in internal units used by the simulator, not SI units.

| | Training | | Testing | |
|---|---|---|---|---|
| **Parameter** | low | high | low | high |
| Mass (kg) | 0.02 | 0.09 | 0.01 | 0.1 |
| $I_{xx}$ (kg · m$^2$) | 2e-6 | 9e-4 | 1e-6 | 1e-3 |
| $I_{yy}$ (kg · m$^2$) | 2e-6 | 9e-4 | 1e-6 | 1e-3 |
| $I_{zz}$ (kg · m$^2$) | 2e-6 | 9e-4 | 1e-6 | 1e-3 |
| $k_\eta$ (N/(rad/s)$^2$) | 2e-8 | 8e-7 | 1e-8 | 1e-6 |

Table 8: Parameter ranges for Quadrotor

were sampled exclusively from the set difference of the test parameter range and training parameter range.

| | Training | | Testing | |
|---|---|---|---|---|
| **Parameter** | low | high | low | high |
| Added Payload (kg) | -0.8 | 2.5 | -1.0 | 4.0 |
| Motor Strength Factor | 0.9 | 1.0 | 0.8 | 1.1 |
| Friction Coefficient | 0.25 | 1.75 | 0.2 | 2.0 |
| Restitution Coefficient | 0.1 | 0.3 | 0.05 | 5.0 |

Table 9: Parameter ranges for Quadruped

# F    Benchmark Function Results

Table 10: Average Final Obtained Value on Benchmark Systems

|  | Average Value over Last 5 Trials ($\downarrow$) | |
|---|---|---|
|  | Branin | Hartmann ($\times 10^{-4}$) |
| F-PACOH [4] | $2.26 \pm 0.70$ | $3.30 \pm 4.55$ |
| Reptile [12] | $3.47 \pm 11.79$ | $\mathbf{1.14 \pm 1.98}$ |
| OCCAM (no-meta) | $1.80 \pm 0.77$ | $7.42 \pm 11.4$ |
| OCCAM (context-only) | $4.25 \pm 3.92$ | $12.83 \pm 15.7$ |
| **OCCAM (Ours)** | $\mathbf{1.65 \pm 0.49}$ | $3.14 \pm 5.97$ |

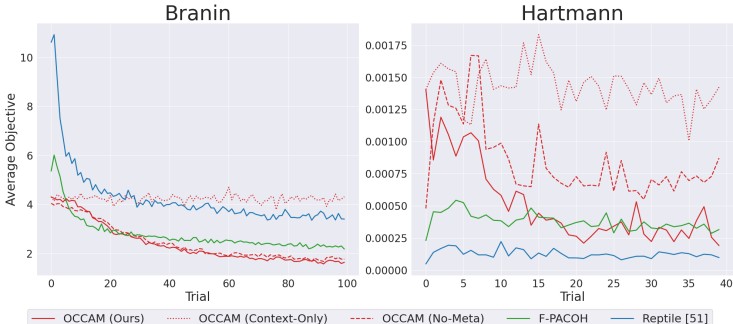

Figure 1: Minima found on each benchmark function (Lower is Better)

We report the average final reward obtained by all methods on the Branin and Hartmann benchmarks in Table 10, and show minima obtained by each method over time in Figure 1. Notably, our method performs well in both settings. In the Branin setting, OCCAM learns a good initialization and finds the best final minimum. In the Hartmann setting, even though OCCAM learns a relatively poor prior, it is able to adapt and find the same final minimum as F-PACOH.

# G    Raw Performance Metrics

Figures 2, 3, and 4 show the raw performance metrics obtained by each method on each system in the trials in which they did not crash. We note that each method is not directly optimizing for these raw metrics, but instead a weighted combination of their normalized versions, so good or bad performance in an individual metric in these plots does not necessarily translate to high or low reward in the plots reported in the paper. For example, in the Racing Car example, our method obtains a lower average speed than many other methods; however, this makes sense as, in the scalarized

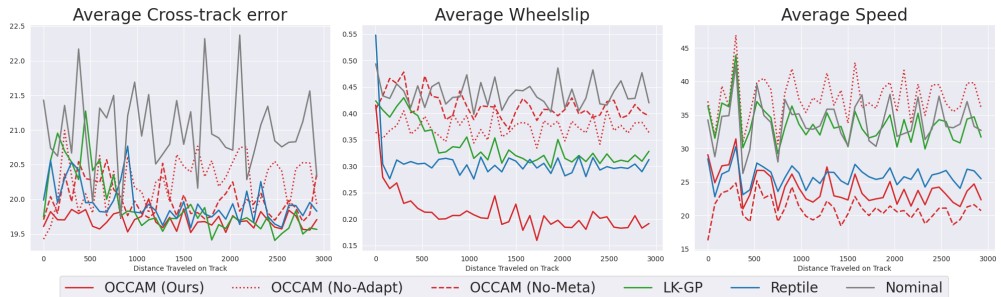

Figure 2: Raw performance metrics obtained by each method on our out-of-distribution racing car test set in successful runs.

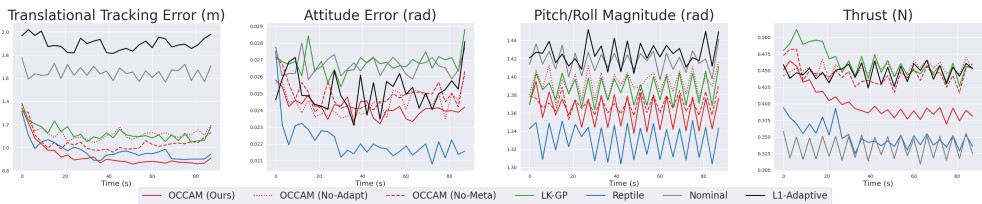

Figure 3: Raw Performance metrics obtained by each method on our out-of-distribution quadrotor test set in successful runs.

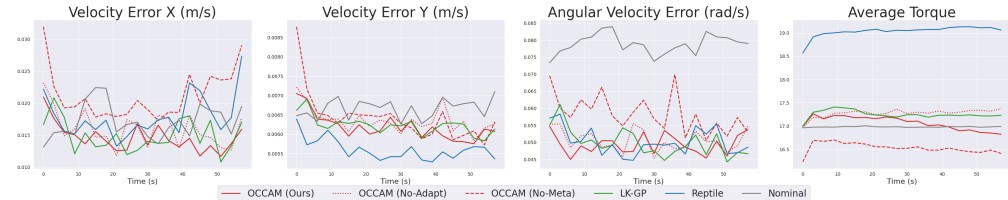

Figure 4: Raw Performance metrics obtained by each method on our out-of-distribution quadruped car test set in successful runs.

objective the model was optimizing for, the speed metric was weighted much lower than the tracking error metric. Also to faithfully report the raw metrics without the crashes skewing the averages, we filter out the runs that crashed. For example, in the quadrotor example, although Reptile performs well when it selects gains that don't result in crashes, its higher crash rate brings down its overall average reward.

## H    Additional Simulation Experiments

### H.1    In-Distribution Experiments

Table 11: Average Final Reward and Crash Rate on In-Distribution Robotic Systems

|  | Race Car | | Quadrotor | | Quadruped | |
|---|---|---|---|---|---|---|
| Method | Avg Final Rwd ($\uparrow$) | Crash % ($\downarrow$) | Avg Final Rwd ($\uparrow$) | Crash % ($\downarrow$) | Avg Final Rwd ($\uparrow$) | Crash % ($\downarrow$) |
| Nominal | $0.50 \pm 0$ | 0 | $1.15 \pm 0.13$ | 47.9 | $0.66 \pm 0.9$ | 14.7 |
| LK-GP | $0.49 \pm 0.08$ | 0 | $1.79 \pm 0.38$ | 37.7 | $0.74 \pm 0.08$ | 8.3 |
| Reptile | $0.42 \pm 0.13$ | 2.7 | $1.19 \pm 0.37$ | 33.8 | $0.72 \pm 0.1$ | 9.4 |
| $\mathcal{L}$1-Adaptive | - | - | $1.37 \pm 0.55$ | 57.5 | - | - |
| OCCAM (context-only) | $0.47 \pm 0.06$ | 2 | $1.94 \pm 0.26$ | 32.5 | $0.76 \pm 0.07$ | 5.6 |
| **OCCAM (Ours)** | $0.44 \pm 0.19$ | 4 | $1.82 \pm 0.40$ | 37.5 | $0.74 \pm 0.09$ | 8.7 |

We also run our method and each baseline on test sets randomly sampled from the training distributions for each of the robotic systems (see Tables 7, 8, and 9). The average final obtained reward and crash rates are reported in Table 11. The performances of each method naturally improve in this setting as the sampled system parameters lie closer to the nominal parameters, but in particular the `context-only` baseline, which only uses the fixed context encoder for sysid, and the LK-GP baseline both obtain amongst the highest rewards and perform similarly to OCCAM, showing, within the training distribution, these approaches perform well.

Also notable in this setting is that the $\mathcal{L}$1-Adaptive controller obtains higher reward than the Nominal controller, demonstrating that the adaptive control does indeed improve performance when the deviation from the nominal dynamics is smaller. However, when the parametric error grows larger in the out-of-distribution experiments in the main paper, the adaptive controller becomes unstable and reduces performance.

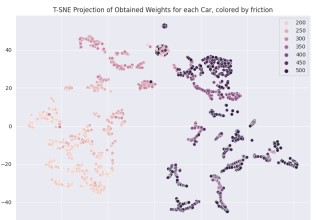

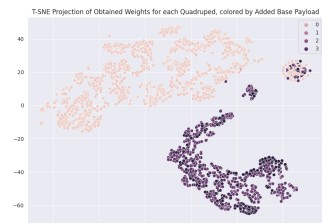

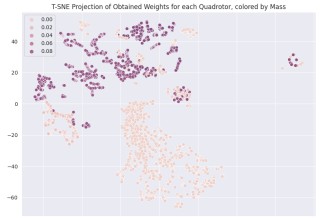

(a) Basis weights computed in the Racing Car Environment, projected into two dimensions and colored by the friction parameter. The weights form distinct clusters separated by different friction coefficients.

(b) Basis weights computed in the Quadruped Environment, projected into two dimensions and colored by the friction parameter. The weights form distinct clusters separated by different added base payloads.

(c) Basis weights computed in the Quadrotor Environment, projected into two dimensions and colored by the friction parameter. The weights form distinct clusters separated by mass parameters

## H.2 OCCAM Makes Interpretable Adaptations to the Gains

To elucidate that our method finds semantically meaningful gains, we run an additional experiment in the racing car environment where we sweep only friction coefficients across 3 different tracks and plot the average final gains chosen by OCCAM in Figure 5. As friction increases, OCCAM selects gains that cause the car to accelerate more aggressively and drive faster, while in the low friction regime, the gains tend towards slower driving (higher brake gain, lower speed in corners). Our method logically chooses a more aggressive driving profile as available traction increases, showing physically meaningful adaptation to changes in system parameters.

## H.3 Is there structure to the learned weight space?

We also include preliminary experiments demonstrating that the space of weights that OCCAM adapts in has meaningful structure. For each test set in the paper, we use t-SNE to project the weights computed by OCCAM's regression procedure into two dimensions and plot the projected weights in Figures 6a, 6c, and 6b. Note that like the weight adaptation procedure, the t-SNE embedding procedure has no knowledge of the underlying system parameters. For each system, the values of the weights distinctly cluster according to the underlying system parameters.

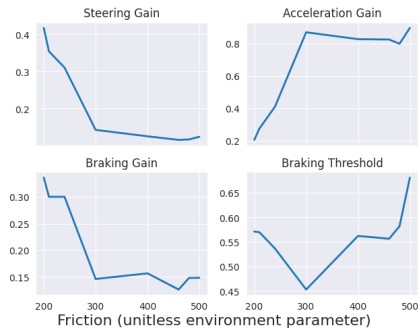

Figure 5: Adapted gains found by our framework for cars with increasing friction coefficients. For cars with higher friction coefficients, our model chooses gains that lead to faster and more aggressive driving. Both the low-end and high-end friction coefficients are out of the training distribution of the model.

## H.4 Gain
## Adaptation with Unseen Environmental Variations

We also run an experiment to assess the ability of our method to tune controllers while novel environmental parameters are varied. We use OCCAM to tune the controller gains of a simulated Crazyflie in three different wind conditions up to 0.5 m/s to minimize the tracking error. Despite the fact that wind was not modeled at all during training, OCCAM is able to achieve an average tracking error of under 7cm, outperforming all baselines, shown in Figure 7. Meanwhile, as expected from the previous section, `no-adapt` fails to tune the controller in this unseen setting.

## I Additional Physical Crazyflie Experiments

We ran additional experiments on the physical Crazyflie platform in which we added a 5-gram mass from the beginning of the experiment and in the middle of the experiment. Plots of the tracking error obtained by the controller with OCCAM's optimized gains, the nominal gains, and with the $\mathcal{L}$1-Adaptive control augmentation are shown in Figures 8a and 8b. In both cases, OCCAM finds gains that result in more robust tracking in the Z-axis. We hypothesize that because our predictive model is trained on data gathered from many quadrotors with varied masses, it learns to select gains that better compensate for these variations.

An interesting result are the minor, high frequency oscillations observed in the Z-axis in Figure 8a and in the X- and Y-axes in Figure 8b towards the end of the experiment. These are most likely the result of marginally

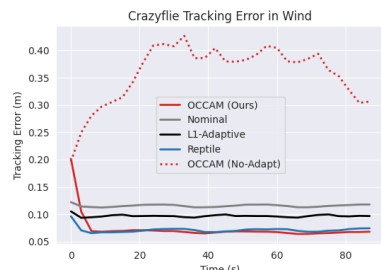

Figure 7: Positional tracking error results on simulated Crazyflie quadrotor following a 3-dimensional ellipsoidal reference trajectory in windy conditions unseen in training. For both versions of OCCAM and Reptile, the control gains are updated every 3 seconds.

stable closed-loop attitude dynamics. One possible solution to this is augmenting the performance measures $y$ and measurement vector $z$ with pitch and roll angular velocities, which might encourage the predictive model and optimizer to select gains that do not result in oscillations. Another solution is to add small random force perturbations to the training simulations so that marginally stable controllers achieve worse performance metrics. We leave exploring these additions to future work.

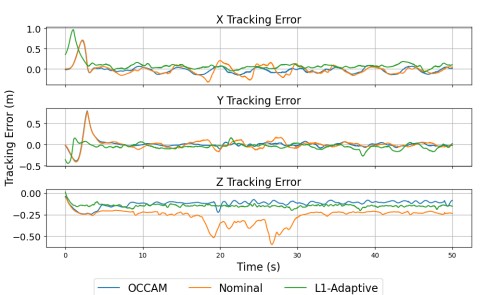 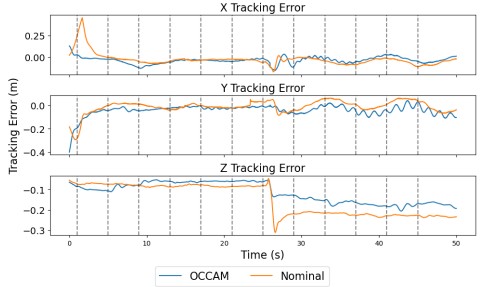

(a) Results with a 5-gram mass added from the start.    (b) Results with a 5-gram mass added at roughly 26s

Figure 8: Positional tracking error results on physical Crazyflie quadrotor following a 3-dimensional ellipsoidal reference trajectory, with added masses.

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
