# OpenReview forum: "OCCAM: Online Continuous Controller Adaptation with Meta-Learned Models"
_robot-learning.org/CoRL/2024/Conference — CoRL 2024_

### Official Review · Reviewer_vgNh · 2024-07-12
**An interesting work that applies meta-learning to controller parameter adaptation. Decision updated on Aug. 15.**

**Originality:** 4
**Technical Quality:** 3
**Clarity Of Presentation:** 3
**Potential Impact:** 3
**Recommendation:** 4
**Confidence:** 5

**Review:**

I think this paper proposed a novel design for controller parameter adaptation for potentially unseen environments or tasks. The presentation was clear, and the results were well presented. However, I have some concerns about the technical details of this paper, which are summarized below.

Major comments/questions

1.	The authors mistakenly categorized L1 adaptive control as direct adaptive control. In fact, it is one type of indirect adaptive control. In the direct architecture, the controller parameters are directly updated, while in the indirect architecture, the plant parameters are estimated and used in the feedback law.  See the references [1,2].

2.	The usage of the Kalman filter (KF) with an encoder output seems to be a novel design. In my understanding, the KF is the key mechanism that introduces “feedback” to the adaptation, where the innovation step in the KF uses the error between the predicted performance measures and actual performance measures, i.e., $y_{t+1} - \Phi w_t$, to update the weights $w_{t+1}$. There are two questions here:

(a) The justification for using KF here is missing. Basically, in classical examples of KF, there’s a dynamic process for which the state can not be measured directly but is observable from noisy sensor measurements. The authors stated that the state in KF, $w_t$ is treated as a normally distributed random variable. But what’s the rationale behind this argument? Some insights will be helpful here to help the readers understand the necessity of KF in the proposed architecture.

(b) Why is $\Phi$ time-invariant (as interpreted by its free of subscript $k$)? The input to the predictive model, including control gains $g_k$, task $\tau_k$, and history control $u_{k-H:k}$ and measurements $z_{k-H:k}$ are all explicitly dependent on time $k$, it is strange that the output of the predictive model $\Phi$ is independent to $k$.

3.	Sections 4.1-4.3 introduce the components of the proposed OCCAM framework. It’d be helpful to state whether they are applied online for adaptation or offline for training, or both. For my current understanding, the control performance modelling (4.1) is applied both in offline training and online adaptation. Meta-training (4.2) is only used in offline training. Controller optimization (4.3) is only used in online adaptation. Is this correct?

4.	In Section 4.2, the authors mentioned that they use backpropagation (BP) to update the network parameters as well as the KF parameters. I understand that BP is the standard tool for updating neural network parameters. However, the KF parameters are usually determined based on the statistics of the sensor’s noise levels but were updated with BP instead. What is the rationale for using BP to update KF parameters here? Is there any physical interpretation?

5.	In Section 4.3, the authors used a modified random-search algorithm to enumerate candidate control parameters and then evaluate them through the performance prediction model to pick the optimal candidate. Maybe better approaches can be taken here to obtain the optimal candidate more efficiently, such as Metropolis-Hastings sampling [3] or policy-search [4,5].

6.	The ablation study clearly shows the contribution of individual components (meta-learning and context encoder) to the generalization of the proposed framework.

7.	Line 250, what do “test parameters” refer to? It’s not clear from the context. Are they the initial parameters used at time = 0 in Figure 3a? Also, what does it mean by “300% parametric error from the nominal model”?

8.	The reported performance on L1 adaptive control seems to contradict the results of the simulation and experiments. In the reported simulation results, the L1AC’s reward and crash rate are close to those of the nominal controller. However, in the experiments, shown in Figure 4, L1AC has a clear advantage over the nominal controller’s tracking performance, which is qualitatively distinctive to what’s been shown in Table 1. The authors may want to double-check the simulation setting to confirm the correctness of the results.

9.	I think the authors should have one paragraph explaining how the "adaptation" is made possible in the current work. My guess for this question is that it is due to the innovation step of the KF. The authors should make it clear because otherwise, this framework simply "guesses" suitable parameters without incorporating feedback from the system regarding the actual performance, which will greatly compromise the quality of the proposed approach.

Minor comments:

1.	The variable $w$ seems typed in two fonts. Please correct it if they indicated the same term.

2.	On top of page 4, the authors stated that the index $t$ is different than $k$, where $t$ is used to denote the time instances for performance measurement and adaptation. If I understand it correctly, the span between $t$ and $t+1$ is longer than that between $k$ and $k+1$. If this is the case, then the authors should make it explicitly clear.

3.	In Section 5.1, for the quadrotor simulation, why is the reward function set to be the linear combination of the inverse of various interpretable metrics like the position tracking error and yaw tracking error? Reciprocals of these interpretable metrics lose the physical meaning carried by these metrics, let alone the linear combination. The authors need to provide justification for this setting.

4.	For reference [13] in the paper, consider citing the more comprehensive work [6] listed below.

References

[1] Hovakimyan, Naira, and Chengyu Cao. ℒ1 adaptive control theory: Guaranteed robustness with fast adaptation. Society for Industrial and Applied Mathematics, 2010.

[2] Hovakimyan, Naira, Chengyu Cao, Evgeny Kharisov, Enric Xargay, and Irene M. Gregory. "L 1 adaptive control for safety-critical systems." IEEE Control Systems Magazine 31, no. 5 (2011): 54-104.

[3] Loquercio, Antonio, Alessandro Saviolo, and Davide Scaramuzza. "Autotune: Controller tuning for high-speed flight." IEEE Robotics and Automation Letters 7, no. 2 (2022): 4432-4439.

[4] Y. Song and D. Scaramuzza, "Policy Search for Model Predictive Control with Application to Agile Drone Flight," IEEE Transaction on Robotics (T-RO), 2021.

[5] Y. Song and D. Scaramuzza, "Learning High-Level Policies for Model Predictive Control," IEEE/RSJ International Conference on Intelligent Robots and Systems (IROS), Las Vegas, 2020.

[6] Cheng, Sheng, Minkyung Kim, Lin Song, Chengyu Yang, Yiquan Jin, Shenlong Wang, and Naira Hovakimyan. "Difftune: Auto-tuning through auto-differentiation." arXiv preprint arXiv:2209.10021 (2024).

Aug. 15: I appreciate the clarification and improvement the authors have made. I have changed my recommendation accordingly.

**Quality Of The Limitations Section:**

3

**Questions For Rebuttal:**

see my numbered questions above.

**Robotics Focus:**

4

**Summary Of Paper:**

This paper proposed an online controller adaptation method to automate the parameter adaptation subject to dynamically changing environments and tasks. The contributions include (1) a prediction model that maps control parameters to their performance in a closed-loop system; (2) the usage of meta-learning to generalize the learned model to different system parameters; and (3) an parameter optimization approach that selects the optimal parameter for deployment. The approach has been validated thoroughly in simulation and experiments over different robotic systems and controllers.

**Summary Of Recommendation:**

The proposed method is novel and the results are relatively comprehensive. The authors should address the concerns listed above. Based on its current state, I recommend this paper to be weakly rejected.

---

### Official Review · Reviewer_27Pw · 2024-07-16

**Originality:** 4
**Technical Quality:** 4
**Clarity Of Presentation:** 3
**Potential Impact:** 3
**Recommendation:** 4
**Confidence:** 3

**Review:**

In my opinion the paper presents an original and important work, and is written clearly.

Strengths:
- interesting approach that combines meta-learning of the basis functions and online estimation of the parameters, that are used to predict the performance measures estimate
- experiments are convincing and done carefully with appropriate ablation studies
- the method is presented quite clearly

Weaknesses:
- contributions of this paper are not expressed clearly, which makes the evaluation of the originality a bit harder
- the timescales of the adaptation seem a bit arbitrary and are relatively low
- minor presentation and notation issues


Issues and comments (not in any particular order, numbers used only for easier reference):

1. It would be nice to have the contributions listed to better understand which parts of the paper are already present in the literature and which ideas are new in the field. For example the idea of using the performance measures estimates to tune the controller is already present for example in the paper "Tuning of extended state observer with neural network-based control performance assessment" which was not referenced.

2. If I understand correctly the \Phi is defined by a neural network, and may change rapidly. How the proposed Kalman filtering scheme for w works in the presence of a rapid change of \Phi, did you observed such behavior?

3. "It is also unclear how a reference model that is always feasible for the actual system to track might be synthesized" This sentence is hard to understand.

4. In algorithm 1:
- "Sample random-size subset D train of size n" random-size or of size n?
- What is G, Z, U and Y?
- Are the datapoints in D_trian correlated (form trajectories) or are sampled at random, and you perform just one step updates?
- How importnat is the optimization of w_0? Is w_0 meant to be common for all datapoints?

5. Why this kind of control parameters choice optimization procedure was used? What is the advantage of the "evolutionary modification to random-search" over different existing optimization techniques?

6. How the \Beta was chosen? How important is the choice of \Beta?

7. In all formulas that optimize for g, g is not present in the objective.

8. The adaptation frequency seems very low, like 1/3 of a lap, 4s and 3s.
Have you tested these methods in the faster changing conditions? What happens in between the adaptations? Is the Kalman filter still running and the estimates are just taken out every \Delta T? What is the frequency of the filtering?

9. The controller used for the race car seems quite unusual. Why this kind of controller was used?

10. " Adaptation with our method also occurs within a few timesteps, which corresponds to 10-20 seconds of data on each system" How this refers to the adaptation frequency \Delta T?

11. Fig 4. why all methods are not starting on the trajectory?
Why the tracking is so bad at the very beginning for nominal and L1?

12. What are the performance measures? What is the impact of the particular choice of performance measures on the resultant average final reward?

13. How well are the performance measures estimated?

14. Why in Fig. 7b (Appendix) there is no comparison with L1?

15. Why do you use [36] in the comparison table and not [37]?

Minor issues:
- Abbreviation "C" for the controller seems to me a bit redundant and not needed.
- "subset Θ" subset of what?

**Quality Of The Limitations Section:**

3

**Questions For Rebuttal:**

I would say that authors should work on fixing the presentation and notation issues, try to answer the questions from the review and introduce a more explicit statement of contributions and explain the adaptation timing better.

**Robotics Focus:**

4

**Summary Of Paper:**

Paper introduces an interesting and effective way of adapting the controller parameters online.

**Summary Of Recommendation:**

In my opinion paper advances an important field of the online controller adaptaiton and shows convincing results.

---

### Official Review · Reviewer_vgNE · 2024-07-21
**Interesting ideas related to meta-training/adapting predictive models to directly predict system performance given a set of control parameters, validated by extensive experiments**

**Originality:** 4
**Technical Quality:** 4
**Clarity Of Presentation:** 4
**Potential Impact:** 3
**Recommendation:** 4
**Confidence:** 4

**Review:**

Combining meta-learning and Bayesian regression (Kalman filter) to update a predictive model to directly predict system performance given a set of control parameters is an interesting idea. Extensive experiments demonstrate the efficacy of the proposed approach.  However,   the rationale of the proposed approach is not sufficiently explained. Additionally, some clarifications are needed.
Strengths:
- Using a predictive model to predict performance dependence on control parameters is an interesting idea.
- Literature review is comprehensive
- Experiments are extensive and involve hardware implementation.

Major Weaknesses:
-  Despite extensive empirical results, it is unclear when the proposed approach can be applied. In other words,  under what conditions (e.g., in terms of dynamics, tasks, performance criteria, etc.) can an good predictive model be learned to directly predict performance dependence on control parameters without identifying the dynamics model?
- The authors propose to use Kalman filter with identity dynamics to estimate the weights $w$, which is “treated as a normally distributed random variable”. It is unclear why this can be done. Also, the Kalman filter is for stochastic dynamics, while the dynamics change from one environment to another is not due to stochasticity.


Aug 16: I appreciate the authors' efforts in addressing my concerns and improving the paper. Most of my concerns have been clarified.  My only remaining concern is that, given a system, it seems difficult to know beforehand whether the method will work well before testing it on that system. I hope this limitation can be addressed in the authors' future work. I updated my rating.

**Quality Of The Limitations Section:**

3

**Questions For Rebuttal:**

1.	Is it possible to explain under what conditions the proposed approach (which relies on a model to directly predict control performance given control parameters) can be applied?
2.	Please justify the use of Kalman filter (see my corresponding comment above).  In addition to explanation, is it possible to do some simulations to show that the proposed approach can almost recover the true weights $w$ (and the basis functions) online under some scenarios (e.g., a scenario that is included in the meta-training stage)
3.	How can the variance $\sigma$ in Section 4.3 be calculated? Also, instead of using “evolutionary modification to random-search”, can the authors use more precise language?
4.	Is the PD controller for the race car a “nonlinear” controller?
5.	In Section 5.3, what is the meaning of “phase-1”?
6.	For the racing car example, the performance improvement with increasing laps seems very little for all methods. What could be the reasons?
7.	In Figure 4, according to the legend, nominal controller gives the best tracking performance. Is this expected?
8.	How the adaptation to different tasks is demonstrated is not straightforward to me. Can the authors elaborate a little bit?

**Robotics Focus:**

4

**Summary Of Paper:**

This paper addresses the challenge of control tuning and adaptation in robots operating across diverse environments. In particular, the authors propose a framework for online controller adaptation that integrates meta-learning with Bayesian recursive estimation. This core idea is to meta-train a predictive models to directly predict system performance given a set of control parameters and online adapt this model using collected data during system operation. The predictive models serve as cost functions in sampling-based optimization routines, enabling the online tuning of control parameters that maximize system performance. The framework’s efficacy is demonstrated on a variety of robotic systems including a real quadrotor.

**Summary Of Recommendation:**

The manuscript contains interesting ideas, which are validated by extensive simulations and hardware experiments.

---

### Author Rebuttal · Authors · 2024-08-07

This upload contains plots corresponding to answer 2b in our "General Response". We observed that the model’s predictive accuracy generally improves and the covariance of the estimate decreases. These plots are shown in the “Rebuttal” file upload (Figures 1 and 2). In the locomotion setting (Figure 3), the model begins with a good prior but is overconfident - in this case, the Kalman Filter appropriately adjusts the covariance to be higher, which is also the expected behavior.

Edit 08/13: We have added preliminary drafts of the revised main paper and supplementary materials.

---

### Decision · Program_Chairs · 2024-09-04

**Decision:**

Accept

**Comment:**

**Post-rebuttal metareview**:

This paper presents a novel approach to online adapt controllers' parameters. In particular, this paper uses meta-learning to train a predictor that predicts the closed-loop controller performance as a function of controller parameters, online fine-tunes such a predictor, and uses this predictor as a surrogate goal to optimize controller parameters. The idea is novel, and experiments are diverse. Most of the reviewers' concerns have been addressed in the rebuttal phase.

--------------------------------
**Pre-rebuttal metareview**:

Strengths:
1. The idea of using meta-learning to train a predictor to predict policy performance as a function of policy parameters is interesting and somewhat novel.
2. Comprehensive literature review.
3. Diverse experiments.

Weakness:
1. Need more analysis about the applicability of the proposed method: in what systems can we accurately predict the policy’s performance?
2. The choice and design of Kalman filter needs more justification.
3. Other issues mentioned by reviewers such as comparing with L1 adaptive control, adaptation timescale, notations, writing, etc.